16S-FASAS: an integrated pipeline for synthetic full-length 16S rRNA gene sequencing data analysis

Zhang Ke 1 2
Lin Rongnan 1 2
Chang Yujun 1 2
Zhou Qing 1 2
Zhang Zhi zhizhang@capitalbio.com 1 2
1 CapitalBio Corporation , Beijing , China
2 National Engineering Research Center for Beijing Biochip Technology , Beijing , China
Moyer Craig
Electronic publication date: 2022 Sep 23
Publication date: 2022
Volume: 10
Electronic Location ID: e14043
Received 2022 Jan 21; Accepted 2022 Aug 21
Copyright: ©2022 Zhang et al.
Copyright year: 2022
Copyright holder: Zhang et al.
License: This is an open access article distributed under the terms of the Creative Commons Attribution License, which permits unrestricted use, distribution, reproduction and adaptation in any medium and for any purpose provided that it is properly attributed. For attribution, the original author(s), title, publication source (PeerJ) and either DOI or URL of the article must be cited.
License URL: https://creativecommons.org/licenses/by/4.0/

Keywords: Metagenome, Full-length 16s rRNA, Taxonomy, Microbiome

Funding: National Key Research and Development Program of China 2020YFC2005406 This work was funded by National Key Research and Development Program of China (No. 2020YFC2005406). The funders had no role in study design, data collection and analysis, decision to publish, or preparation of the manuscript.

==============================
Background

The full-length 16S rRNA sequencing can better improve the taxonomic and phylogenetic resolution compared to the partial 16S rRNA gene sequencing. The 16S-FAS-NGS (16S rRNA full-length amplicon sequencing based on a next-generation sequencing platform) technology can generate high-quality, full-length 16S rRNA gene sequences using short-read sequencers, together with assembly procedures. However there is a lack of a data analysis suite that can help process and analyze the synthetic long read data.

Results

Herein, we developed software named 16S-FASAS (16S full-length amplicon sequencing data analysis software) for 16S-FAS-NGS data analysis, which provided high-fidelity species-level microbiome data. 16S-FASAS consists of data quality control, de novo assembly, annotation, and visualization modules. We verified the performance of 16S-FASAS on both mock and fecal samples. In mock communities, we proved that taxonomy assignment by MegaBLAST had fewer misclassifications and tended to find more low abundance species than the USEARCH-UNOISE3-based classifier, resulting in species-level classification of 85.71% (6/7), 85.71% (6/7), 72.72% (8/11), and 70% (7/10) of the target bacteria. When applied to fecal samples, we found that the 16S-FAS-NGS datasets generated contigs grouped into 60 and 56 species, from which 71.62% (43/60) and 76.79% (43/56) were shared with the Pacbio datasets.

Conclusions

16S-FASAS is a valuable tool that helps researchers process and interpret the results of full-length 16S rRNA gene sequencing. Depending on the full-length amplicon sequencing technology, the 16S-FASAS pipeline enables a more accurate report on the bacterial complexity of microbiome samples. 16S-FASAS is freely available for use at https://github.com/capitalbio-bioinfo/FASAS.

Introduction

16S rRNA gene amplicon sequencing technology is commonly used to determine bacterial taxonomy. At present, most diversity studies on microbial communities are based on sequencing 1–3 highly variable regions (V1–V9) of the 16S rRNA gene (Sirichoat et al., 2020). Partial 16S rRNA gene sequencing is found to be affected by the selection of hypervariable region and the length of reads, and thus it cannot consistently provide valid classification beyond the genus level. Long reads can dramatically widen the genetic field and improve the resolution measured using amplicon sequencing (Phillip et al., 2020). Full-length 16S rRNA gene sequences can be obtained using long-read sequencing technologies (PacBio SMRT sequencing and Oxford Nanopore sequencing) at comparatively high throughput (Santos et al., 2020; Pootakham et al., 2021). Moreover, the PacBio circular consensus sequencing (CCS) technology improves the intrinsic error rate and provides high fidelity species identification (Earl et al., 2018). However, to some extent, the large amounts of input material and high economic cost impede the widespread application of third-generation sequencing (Callahan et al., 2021).

Most synthetic long-read sequencing technology protocols are based on the addition of unique molecular identifiers (UMIs) to the fragmented single long DNA molecules, so that the originating DNA molecules can be reconstructed by assembly with UMI barcodes after sequencing (Chen et al., 2020). Synthetic long-read sequencing technologies are appealing, as they can generate haplotype-resolved genome (Stapleton et al., 2016), full-length transcript (Liu et al., 2021), and full-length 16S rRNA gene sequencing (Dong et al., 2021) data with low-cost and highly accurate next-generation sequencing (NGS) platforms. For instance, Loop Genomics (San Jose, CA, USA) recently has developed a new commercialized technology called loopSeq that reconstructs full-length 16S rRNA gene through de novo assembly combined with the unique molecule barcoding technology (Jeong et al., 2021). Burke and Darling described a method producing high-quality, near full-length 16S rRNA genes sequenced on a short-read sequencer (Burke & Darling, 2016). 16S-FAS-NGS is a similar, low-cost, and high-accurate approach that prepares the linked-tag library and the read-tag library separately before sequencing (Karst et al., 2018). Through the tagging technology, fragmented reads with the same tag are assembled into a single full-length 16S rRNA gene using a de novo assembly algorithm. Before de novo assembly, linked-tag reads are identified and unique tags are extracted, and some unique linked-tag sequences with variants or low abundance are discarded. These challenges are important obstacles to the promotion of 16S-FAS-NGS technology.

The 16S-FAS-NGS is an attractive technology; however, there is a lack of a data analysis suite that can facilitate the assembly, annotation, and visualization of relevant data to help process and analyze the synthetic long read data. Here, we introduce a new tool, called 16S-FASAS, that enables the assembly of the 16S rRNA gene by short reads and subsequent taxonomic composition analysis. The software provides easy-to-use integrated tools for processing 16S-FAS-NGS data.

Materials & Methods

Installation

16S-FASAS is a full-length 16S amplicon sequencing data analysis software that contains modules such as data quality control, sequence demultiplexing, parallel assembly, and taxonomy annotation. Most modules are written in Perl, and an integrated in-shell pipeline is offered, which combines all modules and reads a variety of parameters in the configuration file. 16S-FASAS is hosted on GitHub (https://github.com/capitalbio-bioinfo/FASAS) and can be easily installed locally after downloading the software from the repository. 16S-FASAS does not require administrator privileges to install or run. 16S-FASAS utilizes conda, which provides automatic dependency resolution to install additional software programs and Perl module dependencies. Installation of 16S-FASAS requires the user to simply start with a script named “dep/create_conda_env.sh”, which is created in an isolated conda environment. The output information provides details about installation and reports any errors that occurred. More detailed guidance for implementing 16S-FASAS is available in the README file.

Input

The input data to the 16S-FASAS pipeline consist of raw Illumina sequencing reads from two libraries: a linked-tag library and a read-tag library (Fig. S1). The linked-tag library contains reads with UMIs and flanking primer binding sites, which are used to bin all 16S rRNA gene fragment tag-reads originating from the same parent molecule. The read-tag library contains fragmenting reads with UMIs, which are used to re-create the parent full-length 16S rRNA gene molecules with a de novo assembly algorithm. A configuration file is required for the 16S-FASAS pipeline. The configuration file also records the running parameters and serves as documentation for future reference. Each line in the configuration file represents one parameter for the pipeline.

Architecture

16S-FASAS comprises a set of steps that invoke specific procedures (Fig. 1). Some steps are executed efficiently by taking advantage of parallel computing. 16S-FASAS wraps the execution of these scripts with error-handling code. If the execution of 16S-FASAS is interrupted, the logged error or warning messages help to determine the underlying cause. By default, 16S-FASAS performs the following operations on raw reads in the listed order:

1. Quality control of linked-tag reads. The linked-tag reads consist of adaptor sequences and unique tags. The Hamming distance between the flanking adapters of reads and the true adapter sequences is calculated. Reads are corrected to improve the rate of qualified reads if the hamming distance is less than 3. All reads are qualified with Trimmomatic v0.36, and the two linked-tag sequences are concatenated with XORRO.

2. Extracting unique tags and associated read bins. The unique tags are extracted by identifying the conserved flanking adapters, and the related reads are counted. The unique tag pairs are recorded and sorted by abundance. The tag pairs are used to recruit read-tag reads, thus helping obtain the read bins for each tag pair in the sample. Each bin consists of tag reads originating from the same parent molecule.

3. Quality control of read-tag reads and de novo assembly. Before assembly is performed, all read-tag reads are quality-trimmed and adaptor-trimmed using the Cutadapt software. Then, de novo assembly is implemented on each extracted read bin. The sequencing depths at different regions of a single full-length 16S rRNA gene can be extremely uneven. Two different lightweight algorithms are used by 16S-FASAS for assembling reads with uneven sequencing depths: CAP3 is an Overlap-Layout-Consensus (OLC) assembler, and IDBA-UD is a de Bruijn graph (DBG) assembler. Due to the relatively high resource consumption, assemblers such as spades and megahit are not recommended. Contigs are removed if they are outside the thresholds of the full length of the 16S rRNA gene (those >1.2 kb are retained). All bins are assembled in parallel, and the number of threads is set through the parameters (assemble thread) of the software. The method is chosen through the Assemble Program parameter in the configuration file. All the assembled contigs from each read-tag bin are concatenated into one file and then chimera-filtered using USEARCH 11.

4. Taxonomy assignment of contigs. This module produces abundance tables of contigs that are annotated with their taxonomy using the MegaBLAST tool. The cut-off threshold used to assign taxonomy from MegaBLAST is as follows: (1) alignment length/contig length ≥ 90%, (2) E-value < 1e−20, and (3) identity ≥ 97%, which was performed according to previously published methods (Bolyen et al., 2019). We integrated several frequently used databases for annotation such as SILVA, RDP, and EZbiocloud. The database to be used can be specified in the configuration file. Users can also build their database based on the NCBI Taxonomy database or other microbiome data.

Validation

For validation, the DNA from various microbial species were pooled together to form four mock samples (Table S1). Mock 1 was designed as an in-house mock community that contains a mixture of equal proportions of seven different bacterial species. The Mock 2 community contained gradient proportions of seven organisms. Mock 3 was a more complex in-house mock community that included 11 different organisms. Mock 4 was designed as another in-house mock community, with 10 different species. Mock samples were processed using 16S-FASAS, and species annotation was based on the NCBI Taxonomy database. After quality control and de novo assembly, the contigs were analyzed with two classification methods, MegaBLAST-based classifier and USEARCH-UNOISE3-based classifier, to compare the accuracy of different taxonomic approaches: (1) Abundance tables of contigs were produced using taxonomy assignment module in 16S-FASAS (09.megablast_annotation.pl). (2) Unique contigs were used as input into UNOISE3 algorithm to generate zOTUs (zero-radius Operational Taxonomic Units) in USEARCH (v11.0.667), and then taxonomic classification was performed using the USEARCH sintax command with the NCBI taxonomy database.

Figure 1 Standard steps in the 16S-FASAS pipeline.

Six apparently healthy volunteers from 2017 to 2018 were recruited in this study. From those individuals, fecal samples were collected and used to test the efficiency of 16S-FASAS. Sample collection protocols were performed with the previously published methods (Ma et al., 2018). To compare different full-length approaches, we sent two samples (Fecal 1, Fecal 2) to Novogene (Beijing, China) using PacBio RS II platform for sequence. Full-length 16S rRNA PCR primers were designed as described in a previous study (Karst et al., 2018). Library construction was performed by the Novogene with the Pacific Biosciences Template Prep Kit 2.0. The PacBio dataset was analyzed using the divisive amplicon denoising algorithm 2 (DADA2). Low abundance species (<0.1%) detected were considered as contaminating species, which were excluded from subsequent analysis. All of the visualizations were obtained by using the ggplot2 R package. Raw and assembled sequencing data are available at the NCBI SRA server (https://www.ncbi.nlm.nih.gov/sra/) under project number PRJNA776715.

Results and Discussion

Performance on mock samples

Full-length 16S gene assembly

To estimate the assembly performance of 16S-FASAS, we applied it on four simulated microbial communities with known composition (Mock 1, 2, 3, and 4). Unique tag pairs were extracted from link-tag reads and used for downstream analysis by identifying the known common sequences. Read-tag reads were trimmed, filtered, and binned according to the unique tag pairs. Various indicators of quality control are presented in Table S2. The coverage of the 16S rRNA gene had obvious effects on de novo assembly of full-length 16S rRNA gene sequences. Importantly, 16S-FASAS displayed the distribution of read-tag reads and coverage of the 16S rRNA gene (Fig. 2A). 16S-FASAS filtered contigs by length (>1,200 bp) for downstream analysis. Length distribution of assembled contigs from the mock community is shown in Fig. 2B and Table S3. Some contigs with occasional gaps (N) or less than the expected length were caused by low coverage of reads. The chimera rates of mock samples were 0.12%–0.16% (Table S4). More than 99% of contigs could be identified to species level by MegaBLAST, and the average number of mismatch base pairs was consistent with a previous study (Fig. 2C) (Karst et al., 2018).

Figure 2 Analysis results of 16S-FASAS on mock samples.

(A) Sequencing coverage of mock samples. The X-axis represents the variant region of the 16S rRNA gene. The y-axis represents the number of read 1 (red) and read 2 (blue). (B) The length distribution of mock samples’ contigs. (C) Mismatch distribution from the mock communities. The numbers indicate the percent of all assembled contigs. (D) Comparison of the influence of the classification methods on taxonomic assignment in mock communities. The bar chart represents the relative abundance of species in percentages.

Comparison of classification methods

To further assess the accuracy of species abundance identification, we compared the relative abundance tables generated using MegaBLAST-based classifier and USEARCH-UNOISE3-based classifier (Fig. 2D, Tables S5 and S6). MegaBLAST-based classifier correctly classified six taxa (85.71%, 6/7) into the species level both in Mock 1 and Mock 2, with one species-level discrepancy: classification of Escherichia coli as Escherichia fergusonii and Shigella spp. Shigella spp. is phylogenetically Escherichia spp. and is classified as separate species for medical reasons (Earl et al., 2018). USEARCH-UNOISE3-based classifier correctly identified 85.71% (6/7) and 42.86% (3/7) bacteria at the species level in Mock 1 and Mock 2. In more complex Mock 3, MegaBLAST performed better as well, allowing 72.72% (8/11) of the species to be identified down to the prospective species level. However, the USEARCH-UNOISE3-based classifier performed worse, and only 45.45% (5/11) of the species could be correctly identified. In another more complex Mock 4, we found that 70% (7/10) of target bacteria were correctly identified at the species level when using MegaBLAST-based classifier. In contrast, the USEARCH-UNOISE3-based classifier could classify 60% (6/10) of the target bacteria at the prospective species level. Compared with sintax, MegaBLAST had fewer misclassifications and tended to find more low abundance species, but at the expense of possible false positives. USEARCH showed some trade-offs of accuracy for speed optimization. These results are similar to previous studies (Liber et al., 2021). The possible reasons for the differences observed between MegaBLAST and USEARCH are as follows: (1) USEARCH UNOISE3 is designed for correcting sequencing errors of reads (Edgar, 2016a), which may do not work as well on the assembled contigs. (2) USEARCH sintax is a k-mer based method, which rely on a proxy measurement of the sequence similarity and frequency between the query and reference sequences (Edgar, 2016b) and, therefore, have lower accuracy than sequence alignment in theory (Gao et al., 2017).

For most species, we detected the roughly expected mock taxonomic composition and abundance. However, there were biases observed in the taxonomic profile of mock samples: Klebsiella pneumonia, Haemophilus influenza, and Proteus vulgaris were detected at lower abundances than expected, while an increase in the content of Enterococcus faecium, Streptococcus mutans, and Pseudomonas aeruginosa was observed (Fig. 2D). Two factors might affect the precision of the observed taxonomic abundances: (a) different evolutionary rates of the 16S rRNA gene with multiple copies, and (b) errors induced by experimental conditions, such as DNA extraction, primer esign, and PCR bias. Previous research has provided methods to minimize these effects by tuning the experimental parameters (Burke & Darling, 2016).

Performance on fecal samples

Full-length 16S gene assembly and classification

We performed the same analysis on six fecal samples to verify the applicability of 16S-FASAS. Quality indicators of the fecal samples are summarized in Table S7. Figure 3A shows that the entire variable region of the 16S rRNA gene has high coverage for assembly analysis. Contig assembly statistics of fecal samples are shown in Table S8 and Fig. 3B, and all of their N50 were greater than 1400 bp. Mismatch count distribution for 16S gene sequences from the fecal samples is shown in Fig. 3C. The chimera rates of fecal samples were 0.30%–0.77% (Table S9). We compared the performance of two different taxonomy assignment methods. The results are similar to the performance on mock samples. Most of the species defined by MegaBLAST-based classifier were included in the classification results using the USEARCH-UNOISE3-based classifier. However MegaBLAST-based classifier had higher proportion of assigned contigs than USEARCH-UNOISE3-based classifier at the species level (Tables S10 and S11).

Figure 3 Analysis results of 16S-FASAS on fecal samples.

(A) Sequencing coverage of fecal samples. The X-axis represents the variant region of the 16S gene. The y-axis represents the number of read 1 (red) and read 2 (blue). (B) The length distribution of fecal samples’ contigs. (C) Mismatch distribution from fecal communities. The numbers indicate the percent of all assembled contigs. (D) Venn diagram shows the numbers of unique and shared species between 16S-FASAS and PacBio data sets. (E) Relative abundance analysis of top 30 species in two sequencing methods. Bubble color denote an individual genus, and sizes indicate the relative abundance of an individual species. (F) Memory utilization of the 16S-FASAS on fecal samples. (G) CPU usage of the 16S-FASAS on fecal samples.

Comparison of 16S-FAS-NGS vs. PacBio 16S gene sequencing

To determine whether differences in full-length approaches affected the taxonomic classification, we compared the performance of 16S-FAS-NGS and PacBio sequencing for evaluating microbial community structure on two fecal samples. The 16S-FAS-NGS dataset-generated contigs grouped into 60 and 56 species, of which 28.33% (17/60) and 23.21% (13/56) were unique species. The PacBio sequencing data generated zOTUs grouped into 53 and 58 species, from which 81.13% (43/53) and 74.13% (43/58) were shared with the 16S-FAS-NGS datasets, respectively (Fig. 3D). The relative abundances of the top 30 species are shown in Fig. 3E using the two different sequencing methods. The relative abundance of Megamonas rupellensis, Bacteroides plebeius, and Bacteroides coprocola was high in both the sequencing methods. However, Faecalibacterium prausnitzii was one of the predominant species in 16S-FAS-NGS datasets but was found at low relative abundances in the PacBio sequencing datasets. To some extent, the microbial community profiles represented by 16S-FAS-NGS and PacBio were different. Moreover, we also found some common features in fecal samples using the two sequencing methods. Megamonas rupellensis, Bacteroides plebeius, and Bacteroides coprocola were the dominant species in both sequencing methods. Previous studies have reported that the community profiles using synthetic long-read sequencing technologies (LoopSeq) and PacBio CCS from the same fecal samples were comparable (Yu et al., 2022). Compared to PacBio 16S sequencing, 16S-FAS-NGS offered high fidelity species identification but reduced sequencing prices, which was an attractive technology with species-level resolution.

Computational resources

16S-FASAS was designed to process one sample dataset in a single run. In the part of quality control, assembly, and taxonomy assignment process, 16S-FASAS was implemented using Perl threading module enabled with multi-threading to decrease data processing time. To evaluate the computational resource needs of 16S-FASAS for quality control, assembly, and identification, 16S-FASAS was carried out on six fecal samples. A 16S-FASAS pipeline was run on a Linux workstation (CentOS release 6.5) equipped with Intel(R) Xeon(R) CPU E5-2650 v3 @ 2.30 GHz processors (10 physical cores, 40 threads in total) and 128 GB RAM. We recorded the CPU and memory utilization during analysis to assess the time and resource utilization of 16S-FASAS. The memory (Fig. 3F) and CPU (Fig. 3G) utilization showed two peaks at linked-tag sequence correction and taxonomy annotation, which indicate that quality control and species annotation are two computationally intensive steps.

Conclusions

Obtaining high-quality, full-length 16S rRNA gene sequences based on short reads with molecular tags is a cost-effective technology. Several previous studies have suggested long-read amplicon sequencing of the 16S rRNA gene based on de novo assembly of short Illumina Miseq reads (Karst et al., 2018). However, no mature and easy-to-use software has been available for subsequent analyses. Here, we presented an open-source bioinformatics pipeline called 16S-FASAS that demultiplexes Illumina sequencing data that contain the link and read tags for de novo assembly of the full-length 16S rRNA gene. 16S-FASAS is easy to install, configure, and run. It performs de novo assembly of the full-length 16S rRNA gene with a low error rate through multi-step quality control correction. It generates a species-level relative abundance table through MegaBLAST. 16S-FASAS provides a variety of analysis results and achieves a high degree of automation based on a flexible configuration file. Our results showed that, compared to the PacBio-based method, 16S-FAS-NGS and subsequent 16S-FASAS analyses have similar taxonomic resolution and good price advantage. The good properties and scalability of 16S-FASAS will promote the large-scale application of 16S-FAS-NGS. The application of 16S-FASAS in marker gene sequencing could help refine taxonomic assignments of microbial species and improve the precision of reference databases in future studies.

Supplemental Information

Figure S1 Schematic overview of the 16S-FAS-NGS method

The cDNA molecules of full-length 16S rRNA are uniquely tagged with adaptors containing UMIs (unique molecular tags) and primer binding sites in both ends. The cDNA library are amplified by PCR and split into two parts. The linked-tag library is prepared by circularizing the cDNA molecules to make physical link between left and right UMIs. The read-tag library is prepared by random fragmenting the full-length 16S rRNA genes molecules. These two libraries are pooled and paired-end 150 bp sequenced using the Illumina MiSeq instrument. The direction orders of the paired-end reads are signed with arrow. The linked-tag pairs are used to bin all 16S rRNA gene fragment tag-reads originating from the same parent molecule. The read-tag reads with the same UMIs are used to re-create the parent full-length 16S rRNA gene molecules with a de novo assembly algorithm.

Click here for additional data file.

Table S1 Expected composition and percentage of microbial species in mock samples

Click here for additional data file.

Table S2 Summary of quality indicators of mock samples

Click here for additional data file.

Table S3 Contig assembly statistics of mock samples

Click here for additional data file.

Table S4 Chimera rate of mock samples from USEARCH

Click here for additional data file.

Table S5 Species abundance of mock samples from MegaBlast-based

Click here for additional data file.

Table S6 Species abundance of mock samples from USEARCH-based

Click here for additional data file.

Table S7 Summary of quality indicators of real samples

Click here for additional data file.

Table S8 Contig assembly statistics of real samples

Click here for additional data file.

Table S9 Chimera rate of real samples from USEARCH

Click here for additional data file.

Table S10 Species abundance of real samples from BLAST

Click here for additional data file.

Table S11 Species abundance of real samples from USEARCH-based

Click here for additional data file.

Table S12 Species abundance of real samples from Pacbio sequencing with blast

Click here for additional data file.

Table S13 Species abundance of real samples from Pacbio sequencing with DADA2 (implemented in QIIME 2)

Click here for additional data file.

The authors would like to thank Professor Karst for developing 16S-FAS-NGS technology. We also thank Xiangli Zhang and Yunxiao Ren for giving us several valuable suggestions on data analysis.

Additional Information and Declarations

Competing Interests

Author Contributions

Data Availability

Yujun Chang, Ke Zhang, Rongnan Lin, Qing Zhou and Zhi Zhang are employees of the Capitalbio Corporation, China.

Ke Zhang performed the experiments, analyzed the data, authored or reviewed drafts of the article, and approved the final draft.

Rongnan Lin conceived and designed the experiments, performed the experiments, analyzed the data, prepared figures and/or tables, and approved the final draft.

Yujun Chang conceived and designed the experiments, authored or reviewed drafts of the article, and approved the final draft.

Qing Zhou analyzed the data, prepared figures and/or tables, and approved the final draft.

Zhi Zhang conceived and designed the experiments, authored or reviewed drafts of the article, and approved the final draft.

The following information was supplied regarding data availability:

The data is available at NCBI: PRJNA776715.

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
