# Peer review of "S-FASAS: an integrated pipeline for synthetic full-length 16S rRNA gene sequencing data analysis"

_PeerJ, doi:10.7717/peerj.14043_

## Round 0.1 · original submission · Major Revisions

I agree with both the reviewers in that several critical issues must be resolved before this manuscript can be reconsidered for publication. The primary concern is to clearly distinguish how your pipeline is an improvement to that of Karst et al., 2018 (Retrieval of a million high-quality, full-length microbial 16S and 18S rRNA gene sequences without primer bias). In addition, please attend to each of the comments as carefully outlined by the reviews. You will also need to expand on what “FASAS” refers to as well.

Reviewer 1 ·

Basic reporting

This manuscript describes a software pipeline aimed at analyzing raw data for full-length 16S rRNA gene sequencing data conducted with a method published by Karst et al. (2018).

The pipeline proposed might be very helpful to all of those researchers who like to try out the method proposed by Karst et al. Thus, I am not saying that this pipeline should not be published, but there are several issues in the current manuscript.

Refs seem to be outdated. This field is fast moving and I did not find any Ref from 2020 and more recent. From 2019, I have found 5, while the rest is 2018 or older. This is especially true for Karst et al., 2018 – a seminal publication at the time of publishing, but not anymore.

This manuscript bases its pipeline solely on the method 16S-FAS-NGS by Karst et al., 2018, hence the name 16S-FASAS by the authors. However, from the introduction (and title and abstract) it does not become clear that there is such a specific focus on 16S-FAS-NGS, instead I was under the (first) impression that the manuscript deals with full-length 16S data in general. Only when reading the M&M section, it becomes clear that this manuscript relies on the link-tag and read-tag libraries generated by the method of Karst et al. However, this method (of Karst) seems to be not widely used (correct me). In any case, since the proposed pipeline is very specific to this method, this should be reflected in title, abstract and introduction – otherwise these parts of the manuscript are misleading.

Experimental design

The comparison of full-length 16S is only made towards V4-V5 short amplicons 16S rRNA gene seq. Very obvious, any full-length method will be better compared to any single short amplicon 16S rRNA analysis. You must compare instead to different full-length approaches (PacBio, MinIon, LoopSeq, …) with the method of Karst et al. (or at least more than just V4-V5, which is, of note, inferior compared to V1-V3 or V3-V4 for stool samples anyway).

Unfortunately, my PhD student who has conducted sequencings according to Karst et al. has left the lab. Otherwise, it would have been a pleasure to test the pipeline with our data. I am not able to assess the bioinformatics parts any further.

Concerning mocks, these need to have a certain complexity and must reflect in their species composition the microbiome which is targeted. The chosen bacteria in this manuscript are indeed “various microbial species”, insufficient for stool samples. The performance of such mock communities is not comparable to complex and targeted mocks. Thus, I would think that a mock, which contains at least a dozen bacteria or more and matching the microbiome, should be used. Please have a look at https://doi.org/10.1128/mSphere.01202-20 and https://doi.org/10.3390/microorganisms9061251 for some concepts.

MegaBlast is used for identification here. Well, BLAST is not the best choice of algorithm; there are dedicated tools for 16S rRNA species analysis. The conclusion that MegaBlast is superior above OTU clustering is somewhat misleading, since MegaBlast is an aligner and OTUs are clustered sequences on certain similarity levels. You compare apples with oranges here… Of note, OTUs are ok, but should be phased out and ASVs or zOTUs (denoised amplicons) should be used.

You suggest, beside other databases, using GreenGenes is ok. GG is outdated since 2013 and should not be used anymore (nowhere) and there should be no choice to use GG in order not to tempt the unknowing doctorate student.

Validity of the findings

Shigella are more or less anaerobic Escherichia. Only for historical and medical reasons, they are still two genera.

Concerning the fecal samples, 2 are – well – frugal. Either conduct more or, perhaps even better, use published data and analyze those. Compere the published results with your results.

As already remarked, your study should not just compare full-length to V4V5. Rather delete this unnecessary comparison (outcome was clear before even starting) and broaden your study in adding more full-length samples or re-analyze published data sets as suggested above.

Additional comments

Minor issues
Introduction
First half of the introduction is too general. Focus more about your topic. Add recent studies.
ITS is not a gene.
Materials & Methods: “Haemophilus influenza” and not “Aemophilus”

·

Basic reporting

The manuscript by Chang et al. describes a computational workflow for analyzing 16S rRNA data generated using synthetic long reads. The introduction provides a good level of detail on taxonomic profiling using 16S amplicon sequencing and the caveats associated with it. The authors mention 16S-FAS-NGS technology but do not expand on the abbreviation. It could help a reader to have the abbreviation expanded and detailing how/why this technology is relevant in this context. Additionally it is unclear if 16S-FASAS is also an abbreviation, so expanding what it stands for will be helpful. It is unclear from the introduction that the pipeline is built to analyze synthetic long read data or PCR amplified full length 16S sequences, so adding additional background on what problem the pipeline is intended to solve will be helpful.The methods section lacks a clear description of the workflow used to generate the data. For example, in the methods the authors describe linked-tag libraries and read tag libraries (Lines 97-100). It would be useful for a general audience to have a clear description of what these mean and how the libraries are generated and what read lengths are required for the pipeline .
The description of the computational workflow is a bit sparse, and the manuscript can benefit from additional details for e.g. Line 117 describes tags and flanking adapters. A schematic will be helpful to understand what is being described because from the text it is difficult to follow the methodology. Details on the methods/tools used to extract tags/UMIs will also be good to include.

Line 144 is missing citation for the previously published methods of library prep.
Line 147 it is unclear what the authors mean by the sentence “the contigs were analyzed for operational taxonomic unit (OTU) 148 clustering with QIIME2”. It will be useful to clarify the exact downstream workflow used.

The methods for validation sample collection are not clearly described. On Line 152 the authors mention performing 16S gene V4V5 region sequencing from fecal samples but there is no description of the methods (primers used, lengths of amplicons) or any references to previously described methods. The QIIME2 pipeline offers a large variety of analyses to perform on 16S amplicon data. The authors merely state that they used QIIME2 but fail to provide the details of which set of analyses were carried out and which metrics they used for comparison with their computational workflow to allow for meaningful replication of this work.

In fig 2, the authors describe “quenching reads”. It is unclear what this means as it is not defined anywhere in the manuscript. In fig 3 the caption mentions depth profile at genus and phylum level, however the figure only captures data at the phylum level.
Figure captions can be more detailed as currently it seems insufficient to completely understand the data in the figures, for example it is unclear what is being described in Fig 3B.

Experimental design

The authors have adapted the methodology developed by Karst et al. 2018 (Retrieval of a million high-quality, full-length microbial 16S and 18S rRNA gene sequences without primer bias) and have developed a computational workflow to process data from read tag datasets (synthetic long reads).
Overall, the knowledge gap can be better addressed in the introduction. It is not evident to a reader why this computational workflow should be preferred over other tools and/or the tradeoffs of profiling microbial communities using synthetic long reads vs. amplicon sequencing to highlight the clear benefits of this workflow. It is well known in the microbial genomics community that full length 16S profiling is beneficial for species level classification, but the focus of the manuscript should likely be on what makes this new synthetic long read technology and their associated computational workflow most attractive. It would be useful to address this in the introduction.
Some details could be included in the methods to improve clarity.
The authors use IDBA and cap3 as the assemblers in the workflow. There is little detail provided on the choice of these assemblers and why more common assemblers like spades, megahit which typically perform better are not incorporated in the workflow.
It is also unclear from the description of the computational workflow if the pipeline is designed to run on a single sample vs multiple samples at the same time.
The thresholds used to assign taxonomy from Megablast is not provided. It will be useful to see what these thresholds are and why they were chosen, and how the choice of thresholds contributes to false positives.
In order to validate the workflow, the authors tested the workflow on two mock communities and two fecal microbiome samples. They don’t mention the level of chimeras detected in the data (if any).
The authors describe the workflow run time and CPU usage, but fail to provide specifications/architecture of the system it was run on. Without such specifications runtime information is not meaningful.

Validity of the findings

The manuscript describes a pipeline that uses 16 rRNA synthetic long read data to determine taxonomic composition of microbial communities. While the workflow is not novel from that described in Karst et al, it does provide an automated pipeline to help process and analyze this type of synthetic long read data and provides downstream compatibility with a very commonly used tool for microbial community profiling (QIIME2). Overall, the workflow serves as a wrapper for several tools to possible enable an end user to process their data in a more streamlined manner.

---

## Round 0.2 · Minor Revisions

I agree that the manuscript has been improved, however the focus of the pipeline using the method of Karst et al. (2018) needs to be acknowledged more clearly. Implications for the differences observed (e.g., MegaBlast and USEARCH) should be addressed in the discussion.

Reviewer 1 ·

Basic reporting

Line 6: The short sequencing length of the 16S rRNA gene – That does not make sense. The 16S rRNA gene is about 1.5 kb, irrespective of any sequencing lengths. (also in line 52-53)

Only late in the introduction, the method of KARST et al is mentioned. However, your pipeline (16S-FASAS) seems to be custom build for exactly this type of data produced by the KARST method. Again – mention in Abstract already and not too late in the Introduction.

Introduction between line 29 and line 44 can be deleted, it is too general. Instead, the specific method used to get data for the 16S-FASAS and similar approaches could be introduced (maybe this publication might be useful in this respect: http://www.genome.org/cgi/doi/10.1101/gr.260380.119).

Experimental design

Usearch is a dynamic programing algorithm for fast detection of closest similar sequence. That is why it is used to quickly find the closest match in the reference list. Vsearch performs the Needleman-Wunsch algorithm that is considered better but a bit slower.

Exbiocloud also performs classification using Needleman-Wunsch alignments on their curated list of strains they have, and that is why it is considered the top similarity based classifier. However, using any similarity methods for classification (BLAST, MegaBLAST, Usearch etc) invites all kinds of artifacts, which sole similarity bring and ignore all benefits of positional specific information offers. Similarity based methods should only be used as a tool to extract a bit more information for the relatives of any genus level classified sequence. Thus, positional alignments by dedicated databases should be used.

Validity of the findings

Since the pipeline needs very specific sequencing data (which I do not have), I cannot check the validity of the software.

Additional comments

English still is a bit strange, I am not a native speaker myselft, but language needs to be checked.

·

Basic reporting

The authors were provided detailed feedback on how the manuscript can be improved. While the authors have made some improvements to the narrative, there are still some areas lacking clarity.
1. The methods section jumps from tool installation to library preparation as part of tool inputs, back to architecture, making it incoherent and difficult to follow. Please consider reframing/organizing the workflow better.
2. The grammar can be improved overall in the manuscript, and please fix typos.
3. Figures lack descriptive captions to help the reader understand them better.
4. Venn diagram (Fig 3) is not drawn to scale.
5. The results from the analyses are merely stated and the implications/relevance are not discussed in detail. For example, why the differences observed between MegaBlast and USEARCH are significant and what could cause the difference in observed results.

Experimental design

No comment.

Validity of the findings

From the analysis of the two fecal microbiome samples, the authors showed that synthetic long reads show performance comparable to PacBio.
However, in line 277, the authors state that "results for mock and fecal communities showed that the de novo assembly of the full-length 16s rRNA gene and subsequent MegaBLAST analysis improve the taxonomic and phylogenetic resolution". The results don't necessarily support this conclusion and only show that both methods had similar classification performance.

---

## Round 0.3 · Minor Revisions

I find that the majority of the reviewer suggestions and comments have sufficiently been dealt with. Supplemental figures still lack short descriptive captions to help the reader understand them better and better tie them into the manuscript.

---

## Round 0.4 · Minor Revisions

Please respond to this last minor issue.

Reviewer 1 ·

Basic reporting

Manuscript is ok and can be published.

Experimental design

n/a

Validity of the findings

n/a

·

Basic reporting

No comment

Experimental design

No comment

Validity of the findings

No comment

Additional comments

The authors have sufficiently addressed the comments raised previously. There are some typos in the manuscript that might warrant additional proofreading.
Only comment I have is regarding the captions of Fig 2 and 3. They mention "quenching number of reads" in the caption, however it is not described in the text what they mean by quenching number of reads. It would be good to address/clarify that.

---

## Round 0.5 · accepted · Accept

The issue of explaining what is meant by “quenching number of reads” has been resolved in Figures 2 and 3.